# Early Depletion of Neutrophils Reduces Retinal Inflammation and Neovascularization in Mice with Oxygen-Induced Retinopathy

**DOI:** 10.3390/ijms242115680

**Published:** 2023-10-27

**Authors:** Devy Deliyanti, Varaporn Suphapimol, Phoebe Ang, Xiuying Tang, Abhirup Jayasimhan, Jennifer L. Wilkinson-Berka

**Affiliations:** Department of Anatomy and Physiology, School of Biomedical Sciences, University of Melbourne, Parkville, VIC 3010, Australia; devy.deliyanti@unimelb.edu.au (D.D.); vara.suphapimol@unimelb.edu.au (V.S.); phoebe.ang@unimelb.edu.au (P.A.); xiuyingt@student.unimelb.edu.au (X.T.); abhirup.jayasimhan@unimelb.edu.au (A.J.)

**Keywords:** neutrophils, angiogenesis, vascular leakage, retinopathy

## Abstract

Retinal inflammation is a central feature of ocular neovascular diseases such as diabetic retinopathy and retinopathy of prematurity, but the contribution of neutrophils to this process is not fully understood. We studied oxygen-induced retinopathy (OIR) which develops in two phases, featuring hyperoxia-induced retinal vaso-obliteration in phase I, followed by retinal neovascularization in phase II. As neutrophils are acute responders to tissue damage, we evaluated whether neutrophil depletion with an anti-Ly6G mAb administered in phase I OIR influenced retinal inflammation and vascular injury. Neutrophils were measured in blood and spleen via flow cytometry, and myeloperoxidase, an indicator of neutrophil activity, was evaluated in the retina using Western blotting. Retinal vasculopathy was assessed by quantitating vaso-obliteration, neovascularization, vascular leakage, and VEGF levels. The inflammatory factors, TNF, MCP-1, and ICAM-1 were measured in retina. In the OIR controls, neutrophils were increased in the blood and spleen in phase I but not phase II OIR. In OIR, the anti-Ly6G mAb reduced neutrophils in the blood and spleen, and myeloperoxidase, inflammation, and vasculopathy in the retina. Our findings revealed that the early rise in neutrophils in OIR primes the retina for an inflammatory and angiogenic response that promotes severe damage to the retinal vasculature.

## 1. Introduction

Ocular neovascular diseases such as diabetic retinopathy and retinopathy of prematurity are complex and chronic disorders characterized by the abnormal growth of new blood vessels within the retina. These diseases afflict millions of individuals worldwide and represent a significant public health challenge [1,2,3]. Diabetic retinopathy, often linked to poorly controlled blood sugar levels, stands as a leading cause of blindness in working-age adults [1]. On the other hand, retinopathy of prematurity affects premature infants and is one of the leading causes of blindness in children [3]. Despite their different origins, both diseases share common characteristics, including the development of inflammation and oxidative stress that leads to sight-threatening neovascularization and vascular permeability [4,5,6].

There is growing evidence that some populations of immune cells such as macrophages and CD8^+^ T cells infiltrate the retina through a damaged blood–retinal barrier and secrete cytokines, chemokines, and other factors, which create a microenvironment that promotes retinal vascular disease [7,8]. Identifying the contribution of specific immune cell populations to retinal vasculopathy is crucial for the development of effective treatments. Neutrophils are well known to be rapidly recruited from the circulation as first responders against sites of infection [9,10]. They are also involved in many non-infectious conditions, promoting inflammation in atherosclerosis, arthritis, and acute lung infection [11,12,13]. Additionally, neutrophils increase oxidative stress in a process known as neutrophil extracellular trap (NET) via the secretion of proteases such as myeloperoxidase (MPO), a producer of powerful oxidants that damage the vascular endothelium [14]. Although not fully understood, there is evidence that NET formation occurs in the eye [15,16,17]. Studies in murine diabetic retinopathy [15,16,18,19,20,21] indicate that neutrophils promote inflammation and vascular injury, and in patients with diabetic retinopathy increased neutrophil counts and the presence of NET components suggest the involvement of neutrophils in retinal capillary degeneration [22]. Despite these findings, the precise contribution of neutrophils to retinal neovascularization remains unclear. Our study aimed to determine whether neutrophils influence the development of retinal neovascularization by investigating a mouse model of retinopathy of prematurity known as oxygen-induced retinopathy (OIR).

## 2. Results

### 2.1. Neutrophils Are Increased in Phase I but Not Phase II OIR

OIR develops in two phases [23]. In phase I, the exposure of neonatal mice at postnatal day (P) 7 to hyperoxia mimics the clinical situation in which some preterm infants receive supplemental oxygen to assist breathing, which results in extensive retinal vaso-obliteration. At P12, the exposure of mice to room air for approximately one week comprises phase II OIR, which induces retinal ischemia and the excessive production of angiogenic factors that cause marked neovascularization and vascular leakage in the inner retina [23]. Neutrophils are the first responders to tissue inflammation [10]; therefore, we evaluated their frequency in phase I OIR in the blood and spleen as these are potential sources of neutrophils (Figure 1A). The number of neutrophils in both the blood and spleen was increased in the OIR controls in phase I at P9 compared to age-matched room air controls, but was not increased at the end of phase II OIR at P18 (Figure 1B,C).

### 2.2. Neutrophil Depletion Reduced Neutrophils in the Blood and Spleen and Retinal Vaso-Obliteration in OIR Mice

We next evaluated whether the depletion of neutrophils in phase I OIR with an anti-Ly6G mAb reduced neutrophils in the blood and spleen. The anti-Ly6G mAb or isotype control IgG2a mAb was administered at the onset of OIR at P7 and then every third day until P18 (Figure 2A and Figure 3A). At P12 OIR, the anti-Ly6G mAb reduced neutrophils in the blood and spleen of OIR mice by over 95% compared to OIR mice administered the IgG2a control mAb (Figure 2B). Vaso-obliteration peaks in phase I at P12, and is an important predictor of neovascularization in phase II of the disease [23]. As expected, vaso-obliteration developed in the central retina of OIR mice administered the isotype control IgG2a mAb (Figure 2C,D). The neutrophil depletion antibody reduced retinal vaso-obliteration compared to OIR + IgG2a at P12 (Figure 2C,D).

### 2.3. Neutrophil Depletion Reduced Myeloperoxidase and VEGF in the Retina of Early Phase II OIR

The relative tissue hypoxia that occurs at the commencement of phase II OIR due to mice transitioning from a hyperoxic to normoxic environment stimulates an intense inflammatory and angiogenic response that is involved in the subsequent development of retinal neovascularization (Figure 3A). We therefore measured the activity of MPO, an enzyme that is predominately expressed in neutrophils and whose levels reflect neutrophil activity [24]. OIR mice were treated in phase I and II OIR (Figure 3A). At P13, MPO expression was increased in OIR mice administered the isotype control IgG2a mAb compared to room air controls (Figure 3B and Appendix A). In OIR mice, the anti-Ly6G mAb reduced the expression of MPO in the retina at P13 (Figure 3B and Appendix A). 

Vascular endothelial growth factor (VEGF) is a potent angiogenic, vascular permeability, and inflammatory mediator in ocular neovascular diseases including OIR [25]. At P13, VEGF protein levels in the retina were increased in OIR + IgG2a compared to room air controls (Figure 3C). In OIR mice treated with the anti-Ly6G mAb, VEGF protein expression in the retina was reduced compared to OIR + IgG2a (Figure 3C).

### 2.4. Neutrophil Depletion Reduced Inflammatory Factors in the Retina of Phase II OIR

We next examined whether neutrophil depletion reduced the expression of inflammatory factors involved in the retinal neovascularization that is established by P18 (Figure 3A). In the retina of OIR + IgG2a at P18, the mRNA levels of tumor necrosis factor (*TNF*) and intercellular adhesion molecule-1 (*ICAM-1*) were increased compared to room air controls (Figure 4A,B). Treatment of OIR mice with the anti-Ly6G mAb reduced the mRNA levels of *TNF* and *ICAM-1* to control levels (Figure 4A,B). Similarly, in retina of OIR mice at P18, the increase in monocyte chemoattractant protein-1 (MCP-1) protein levels was reduced in mice treated with the anti-Ly6G mAb (Figure 4C).

### 2.5. Neutrophil Depletion Reduced Retinal Vasculopathy in Phase II OIR

In OIR mice administered the isotype control IgG2a mAb, neovascularization and vasco-obliteration were present in the central retina (Figure 5A–D). In OIR mice treated with the anti-Ly6G mAb, retinal neovascularization was reduced and retinal vaso-obliteration was unchanged compared to the OIR + IgG2a group (Figure 5A–F). In the retina, VEGF protein levels and vascular leakage were increased in OIR + IgG2a compared to room air controls (Figure 5F,G). In OIR mice treated with the anti-Ly6G mAb, both VEGF protein levels and vascular leakage in the retina were reduced but not to the level of room air controls (Figure 5G,H).

### 2.6. Body Weight

As previously reported [7,26], the body weight of OIR mouse pups was lower than age-matched control mice exposed to room air (Appendix A). Treatment of OIR mice with the anti-Ly6G mAb commencing at P7 did not influence body weight compared to OIR mice administered the isotype control IgG2a mAb (Appendix A).

## 3. Discussion

The major findings from this study are the ability of neutrophils to promote retinal inflammation and vasculopathy, including vision-threatening neovascularization and vascular leakage. We demonstrate a significant increase in the number of neutrophils in the blood and spleens of mice during phase I but not late phase II OIR, indicating that neutrophils rapidly respond to the hyperoxia stimulated by the exposure of mice to a high oxygen environment. However, it is possible that neutrophils are increased in the early phase II of OIR when the retina becomes relatively hypoxic, and the process of neovascularization commences. We report a vasculo-protective effect of neutrophil depletion when administered during the commencement of OIR, which includes a reduction of vaso-obliteration in phase I and extends to neovascularization and vascular leakage in phase II. The mechanisms underpinning these benefits of neutrophil depletion imply the presence of neutrophils in the retina. Although this was not directly established, neutrophils are a major source of MPO and therefore the reduced levels of MPO in the retina following neutrophil depletion suggests their presence in retinal tissue. The effects of neutrophil depletion can be attributed to the reduction in inflammatory factors in the OIR retina. It could be speculated that because neutrophils are powerful stimulators of inflammatory pathways, their almost complete absence following neutrophil depletion suggests that they are a major source of these injurious factors in OIR. A limitation of our study is that we did not establish whether neutrophils produce these inflammatory factors, and it is likely that other immune cell populations such as CD8^+^ T cells, microglia, and macrophages are involved [7,27,28]. Nevertheless, our findings provide insights into the role of neutrophils in the development of OIR which may have implications for the pathogenesis of vascular inflammatory disorders of the retina and potential treatment strategies.

The retinal hyperoxia that occurs in phase I OIR features vascular instability that ultimately results in the demise of endothelial cells [23]. In our study, we observed a reduction in vaso-obliteration in mice with OIR following neutrophil depletion, indicating the significant role played by neutrophils in the loss of retinal vessels. To our knowledge, this function of neutrophils in the degeneration of retinal blood vessels has not previously been explored in OIR. Apart from OIR and retinopathy of prematurity, vaso-obliteration is relevant to diabetic retinopathy, which can feature the progressive loss of retinal capillaries (pericytes and endothelial cells) resulting in tissue ischemia [29,30]. Studies on murine diabetic retinopathy have reported a decrease in retinal capillary cell death when neutrophil-associated inflammatory mediators and their activity were ablated [15,18]. Our findings align with these prior investigations, suggesting the involvement of neutrophils in mediating the loss of the retinal vasculature in phase I OIR.

The relative tissue hypoxia in the early phase II of OIR stimulates an oxidative stress, angiogenic, and inflammatory environment within the retina [23,31]. Neutrophils are a central source of MPO, an enzyme that is a potent inducer of oxidative stress and pro-inflammatory mediators [32,33]. It is established that oxidative stress contributes to vascular pathology in the retina in OIR and other neovascular retinopathies such as diabetic retinopathy [34,35,36]. In the present study, the expression of MPO in the retina was increased at P13 in early phase II OIR and reduced with the anti-Ly6G mAb, indicating the importance of neutrophils in promoting a pro-inflammatory and pro-oxidative environment in the retina. The increase in MPO at P13 was accompanied by elevated levels of VEGF, a potent angiogenic factor whose elevated expression is an established response to tissue hypoxia and a compensatory mechanism to revascularize the vaso-obliterated retina. However, the sustained increase in retinal VEGF leads to pathological neovascularization and the vascular permeability that characterises the later stages of phase II OIR [31,37]. The mechanism involved in the decrease in VEGF following neutrophil depletion in OIR is not completely clear. One possibility is that reduced vaso-obliteration in phase I OIR led to less retinal hypoxia in phase II OIR and, thereby, an attenuated stimulus for VEGF production. Alternatively, although VEGF is mainly produced by glial cells and ganglion cells in the OIR retina [38], neutrophils can produce VEGF [39,40], although further research is required to confirm if neutrophils are a source of VEGF in OIR. 

In ocular neovascular diseases, inflammation plays a crucial role by causing vascular damage and promoting abnormal blood vessel growth [7,26]. While VEGF has been traditionally considered the primary mediator of retinal neovascularization, recent studies have highlighted the contribution of inflammatory mediators such as TNF, ICAM-1, and MCP-1 to this process [27,30,41,42]. Neutrophils are known to release a variety of inflammatory mediators, including TNF and MCP-1 [9,43], which can induce oxidative stress and tissue damage, and promote an inflammatory microenvironment, all of which significantly contribute to the development of neovascularization in OIR [27,41,44]. In our study, we found a reduction in retinal *TNF* and MCP-1 levels when neutrophils were depleted. These findings underscore the contribution of neutrophils to the inflammatory response in OIR and suggest that targeted neutrophil depletion may hold promise as a therapeutic approach for reducing retinal inflammation in ocular neovascular diseases.

A major finding in the present study was the reduction in retinal neovascularization and vascular permeability due to neutrophil depletion. This finding is in agreement with prior studies, demonstrating that pathological angiogenesis is induced by neutrophils in models of choroidal [45], corneal [40], and tumor angiogenesis [46,47]. Nonetheless, the biphasic nature of OIR warrants a closer evaluation of the angiogenic response to neutrophil depletion. A recent elegant study by Binet et al. [48] evaluated neutrophil depletion in murine OIR, focusing on the regression of diseased vessels that naturally occurs at the conclusion of retinal neovascularization (approximately post-P17). An interesting finding was that neutrophil depletion with the anti-Ly6G mAb administered at P16 impaired the clearance of diseased endothelial cells and the remodelling of unhealthy vessels. The results of this research, placed in the context of our study, suggest that neutrophils have a dual role in OIR: early depletion in phase I reduces retinal oxidative stress and inflammation and thereby attenuates vaso-obliteration and neovascularization, while late depletion interrupts vascular remodelling and prolongs retinal vasculopathy. Thus, neutrophils have a complex involvement in OIR, damaging the vasculature in early OIR and protecting the vasculature in the remodelling stage of the disease. 

Overall, our findings highlight the crucial role of neutrophils in the development of OIR. These immune cells actively contribute to both vaso-obliteration and neovascularization, as well as the production of inflammatory and angiogenic factors. Our study suggests that targeting neutrophils could be a potential therapeutic strategy for the treatment of ocular neovascular diseases. Further research is required to investigate the precise mechanisms through which neutrophils exert their effects and to explore potential interventions aimed at modulating their activity.

## 4. Materials and Methods

### 4.1. Animals

Litters of C57BL/6J mice were purchased from the Australian Resource Center (Perth, Western Australia). Both male and female mouse pups and their nursing mothers were included in the study. The mice were housed in a temperature-controlled room with a 12 h light/dark cycle and had access to food and water ad libitum. All animal procedures conducted in this study were approved by the University of Melbourne School of Biomedical Sciences’ Animal Ethics Committee (number 20254). 

OIR was performed according to our previous publications [7,26]. OIR consists of two phases. In phase I, between P7 and P12, mouse pups and their nursing mothers were exposed to hyperoxia (75% oxygen) for 22 h per day in specialized chambers (Figure 1A). The oxygen concentration in the chambers was controlled using a ProOx 110 gas regulator (Biospherix, Parish, NY, USA) and attached to medical-grade oxygen cylinders (Air Liquide, Victoria, Australia). Phase II of OIR began when the mice were housed in room air from P12 until P18 (Figure 1A). This methodology complies with our institutional animal ethics committee regulations which require the litters of mice to be exposed to a short period of room air each day during phase I to prevent any respiratory distress that may occur in the mothers. We found there were no differences in the retinal pathology that occurs from this protocol compared to continuous exposure to hyperoxia [23]. Phase II has a slight modification to methodologies in which mice are killed on P17 [23]. We found no differences in retinal pathological or physiological angiogenesis between P17 and P18 [7,26]. Mice were randomized to room air control and OIR control groups. The OIR group was then randomized to treatment with the anti-Ly6G mAb or isotype control IgG2a mAb. The anti-Ly6G mAb (BioX Cell, clone 1A8, #BE0075-1, Lebanon, NH, USA) was administrated at a dose of 100 μg per mouse in 50 μL of 0.1 M phosphate-buffered saline once every 3 days from P7 until P18 via intraperitoneal injection, according to previous studies [49,50]. The isotype-matched control IgG2a mAb (BioXCell, IgG2a, clone 2A3, #BE0089) was administered at the same dose and schedule as the anti-Ly6G mAb. The body weight of pups was recorded throughout the study. At P12 and P18, mice were humanely euthanized with sodium pentobarbitone (170 mg/mL, Virbac, Milperra, NSW, Australia) and their blood, spleen, and retina were collected. Age-matched controls were housed in room air (21% oxygen).

### 4.2. Flow Cytometry

Blood and spleen samples were prepared as previously described [7,26]. Briefly, spleens were harvested and mechanically disrupted using a 40 μm cell strainer to generate single-cell suspensions. Blood and spleen cells were then treated with 1× RBC lysis buffer (#00-4333-57, Thermo Fisher Scientific, Scoresby, VIC, Australia) to remove red blood cells. One million cells were resuspended in 100 μL of flow cytometry buffer (0.1 M phosphate-buffered saline containing 2% fetal bovine serum and 0.1% sodium azide) and incubated with a cocktail of fluorescently labeled antibodies against CD45 (BV786, BioLegend, #304048, San Diego, CA, USA), CD11b (AF700, BioLegend, #101222), and Ly6G (PE-eFluor610, Thermo Fisher Scientific, #61-9668-82) for 45 min at 4 °C in the dark. Antibody concentrations were optimized according to the manufacturer’s recommendations. After incubation, cells were washed with flow cytometry buffer containing 1% fetal bovine serum and 0.5 mM EDTA in 0.1M phosphate-buffered saline and analyzed using a flow cytometer (Fortessa X-20, BD, Biosciences, San Jose, CA, USA). Data analysis was performed using FlowJo software (v10, Tree Star, Inc., Ashland, OR, USA).

### 4.3. Retinal Vaso-Obliteration and Neovascularization

The eyes from mouse pups at P12 were enucleated and immediately fixed in 4% paraformaldehyde for 30 min at room temperature. The retinas were then carefully dissected under a dissecting microscope and stained with fluorescein-labeled isolectin GS-IB4 (Vector Laboratories, Burlingame, CA, USA) to visualize the vasculature. Images were captured using a fluorescence microscope (Zeiss Axio, Zeiss, Jena, Germany) attached to a camera (AxioCam MRc, Carl Zeiss, Gottingen, Germany), and the areas of vaso-obliteration were quantified using ImageJ software v1.5.3 (National Institute of Health, Bethesda, MD, USA). The percentage of vaso-obliteration was calculated as the ratio of the avascular area to the total retinal area, according to our previous studies [26,51]. At P18, the eyes from mouse pups were enucleated and fixed in 4% paraformaldehyde for 30 min at room temperature. The retinas were dissected, and the neovascularization was assessed via staining with isolectin GS-IB4. The neovascular tufts were quantified using ImageJ software v1.5.3 (National Institute of Health, Bethesda, MD, USA) according to our previous publications [7,26].

### 4.4. ELISA Assays for Vascular Leakage, VEGF, and MCP-1

ELISA assays were performed on retinal lysates. Enucleated eyes were immediately dissected, and retinas were snap-frozen in liquid nitrogen until use. Retinal tissues were homogenized in TE-PER lysis buffer (Thermo Fisher, #87510), which was supplemented with a protease and phosphate cocktail mix (Thermo Fisher, #78440) to enhance protein stability and prevent degradation. Total retinal protein concentration was measured using a Bradford kit (Bio-Rad, Hercules, CA, USA) to ensure equal loading of protein samples. ELISA assays were performed according to the manufacturer’s instructions using VEGF (#DY564, R&D Systems), albumin (Immunology Consultants Laboratory, #E-90AL), and MCP-1 (R&D systems, #MJE00B) ELISA kits. Briefly, retinal lysates were added to pre-coated ELISA plates and incubated with primary antibodies specific for VEGF, albumin, or MCP-1. After washing, the plates were incubated with secondary antibodies conjugated to horseradish peroxidase (HRP). HRP activity was detected using a chromogenic substrate, and the absorbance was measured at 450 nm using a microplate reader (FLUOstar Omega, BMG Labtech, Mornington, VIC, Australia).

### 4.5. Western Blotting

MPO protein expression was detected via Western blotting. Briefly, retinal lysates were prepared using RIPA buffer (Thermo Fisher Scientific, #89900) supplemented with protease and phosphatase inhibitors (Thermo Fisher Scientific, #78440). Protein concentration was determined via the Bradford assay (Thermo Fisher Scientific, #23225). Equal amounts of protein (20 μg) were separated on a 10% SDS-PAGE gel (NuPAGE 10% Bis-Tris, ThermoFisher Scientific, #NP0330BOX) and transferred onto a nitrocellulose membrane (Bio-Rad, #1620112, Hercules, CA, USA). The membrane was blocked with 5% non-fat milk in TBST (Tris-buffered saline containing 0.1% Tween-20) for 1 h at room temperature and then incubated with a primary antibody against MPO (Abcam, #ab139748, Cambridge, UK) at a dilution of 1:1000 overnight at 4 °C. After washing with TBST, the membrane was incubated with a horseradish peroxidase (HRP)-conjugated secondary antibody (Cell Signaling Technology, #7074) at a dilution of 1:2000 for 1 h at room temperature. Protein bands were visualized using the ECL detection system (Bio-Rad, #1705060) and a ChemiDoc XRS^+^ (BioRad). Signals were quantified using ImageJ software v1.5.3 (National Institute of Health, Bethesda, MD, USA) and the expression levels of MPO were normalized to the expression levels of α-tubulin (Cell Signaling Technology, #3873, Beverly, MA, USA) as a loading control.

### 4.6. Quantitative Real-Time PCR

The expression levels of *VEGF*, *TNF*, and *ICAM-1* were analyzed. Briefly, total RNA was extracted from retina using a RNAeasy mini kit (#74104, Qiagen Pty Ltd., Clayton, VIC, Australia) according to the manufacturer’s instructions. RNA concentration and purity were determined using a NanoDrop spectrophotometer (Thermo Fisher Scientific). cDNA was synthesized from 1 μg of total RNA using the High-Capacity cDNA Reverse Transcription Kit (Thermo Fisher Scientific, #4368814) according to the manufacturer’s instructions. The qPCR reaction was performed using the PowerUp SYBR Green Master Mix (Thermo Fisher Scientific, #A25742) on a StepOnePlus Real-Time PCR System (Thermo Fisher Scientific). The primer sequences for *ICAM-1*, and *TNF* are published elsewhere [7,26,51]. The expression of mRNA levels was normalized to the expression levels of the housekeeping gene 18s using the 2^−ΔΔCt^ method. 

### 4.7. Statistics

All data were analyzed using GraphPad Prism version 8.0 (GraphPad Software, La Jolla, CA, USA). Normality of the data was assessed using the Shapiro–Wilk test. If the data were normally distributed, the results were presented as mean ± standard deviation (SD) and were analyzed using Student’s *t*-test for two-group comparisons or one-way analysis of variance (ANOVA) followed by Tukey’s post hoc test for multiple-group comparisons. If the data were not normally distributed, the results were presented as median with interquartile range (IQR) and were analyzed using the Mann–Whitney U test for two-group comparisons or the Kruskal–Wallis test followed by Dunn’s post hoc test for multiple-group comparisons. Statistical significance was set at *p* < 0.05.

## Figures and Tables

**Figure 1 ijms-24-15680-f001:**
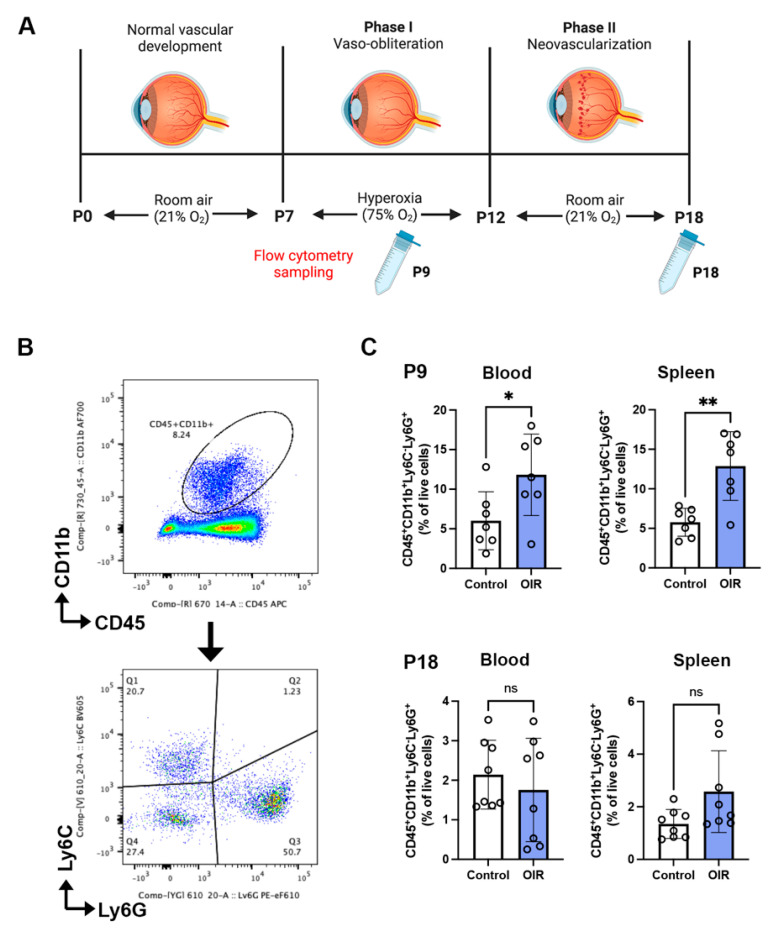
The anti-Ly6G mAb increased neutrophils in the blood and spleen of C57BL/6J mice during phase I OIR (**A**). Schematic diagram illustrating the OIR model and sampling for flow cytometry (**B**). Gating of CD45+ CD11b and Ly6C+Ly6G+ neutrophils in room air control and OIR control mice, and their frequency in the blood and spleen at (**C**) postnatal (P) 9 and P18. * *p* < 0.05, ** *p* < 0.01, and ns, non-significant according to unpaired *t*-test. *n* = 7 mice per group from two independent experiments. All values are mean ± SEM.

**Figure 2 ijms-24-15680-f002:**
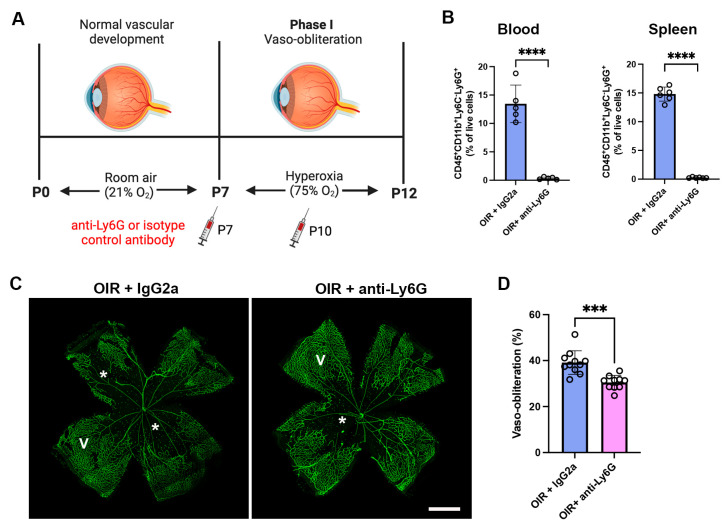
The anti-Ly6G mAb reduced neutrophils in the blood and spleen of C57BL/6J mice in phase I OIR (**A**). Schematic diagram showing the administration of the isotype control IgG2a mAb and the neutrophil depleting anti-Ly6G mAb during phase I OIR (**B**). The frequency of neutrophils in the blood and spleen at P12. **** *p* < 0.0001 according to unpaired *t*-test. *n* = 6 to 7 mice per group from two independent experiments. (**C**) Representative retinal flat-mount images from OIR mice at P12, stained with fluorescein isothiocyanate-conjugated isolectin (green) to visualize blood vessels. Vaso-obliterated (asterisk) and vascularized (V) areas are shown. Scale bar = 150 µm. (**D**). Quantitation of vaso-obliterated area. *** *p* < 0.001 according to unpaired *t*-test. *n* = 11 to 2 mice per group from three independent experiments. All values are mean ± SEM.

**Figure 3 ijms-24-15680-f003:**
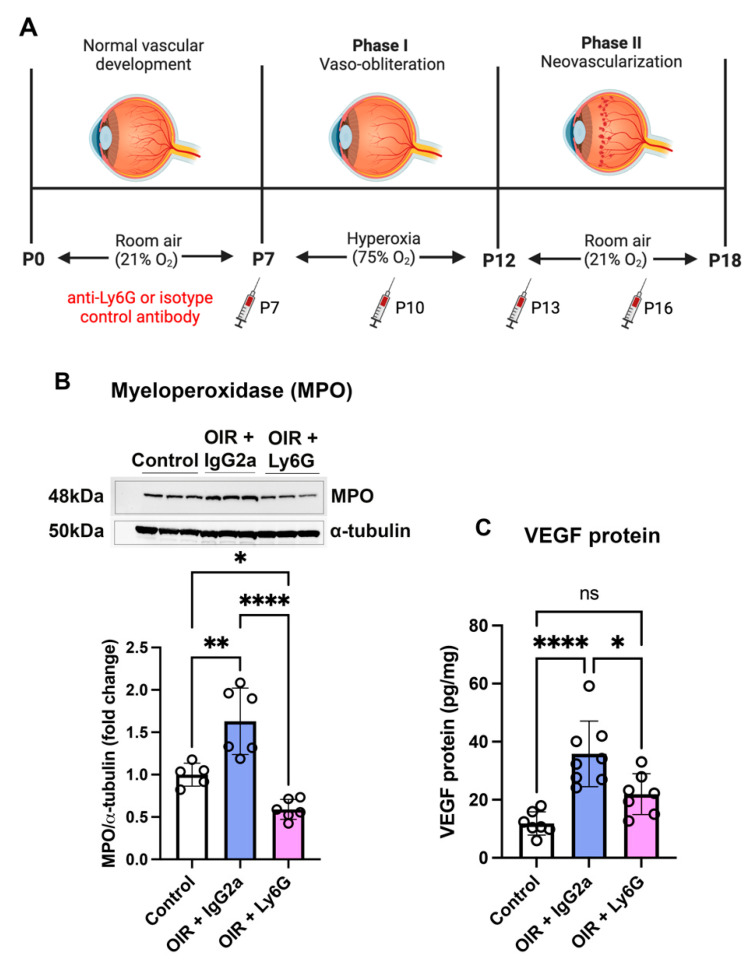
Neutrophil depletion reduced MPO and VEGF levels in the retina of C57BL/6J mice in early phase II OIR (**A**). Schematic diagram showing the administration of the isotype control IgG2a mAb and the neutrophil depleting anti-Ly6G mAb during phase I and II OIR (**B**). MPO expression via Western blotting at P13 (**C**). VEGF protein levels via ELISA at P13. * *p* < 0.05, ** *p* < 0.01, **** *p* < 0.0001, and ns, non-significant according to one-way ANOVA followed by Tukey’s multiple comparisons test. *n* = 6 to 7 mice per group. All values are mean ± SEM.

**Figure 4 ijms-24-15680-f004:**
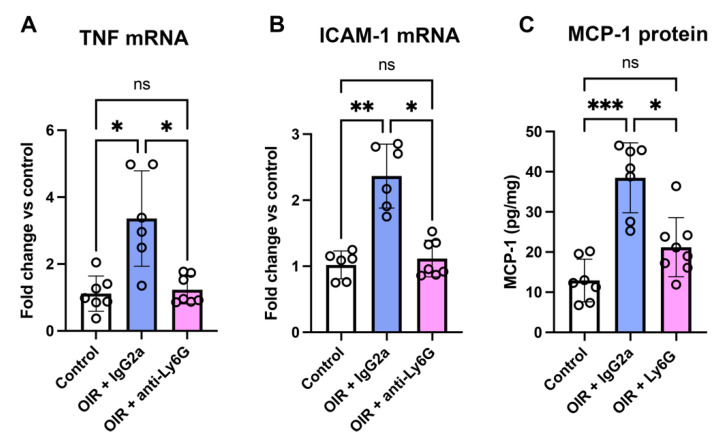
Neutrophil depletion reduced retinal inflammation in OIR. mRNA levels of (**A**) *TNF* and (**B**) *ICAM-1*, and protein levels of (**C**) MCP-1 in retina at P18. * *p* < 0.05, ** *p* < 0.01, *** *p* < 0.001, and ns, non-significant according to one-way ANOVA followed by Kruskal–Wallis test. *n* = 6 to 7 mice per group. All values are mean ± SEM.

**Figure 5 ijms-24-15680-f005:**
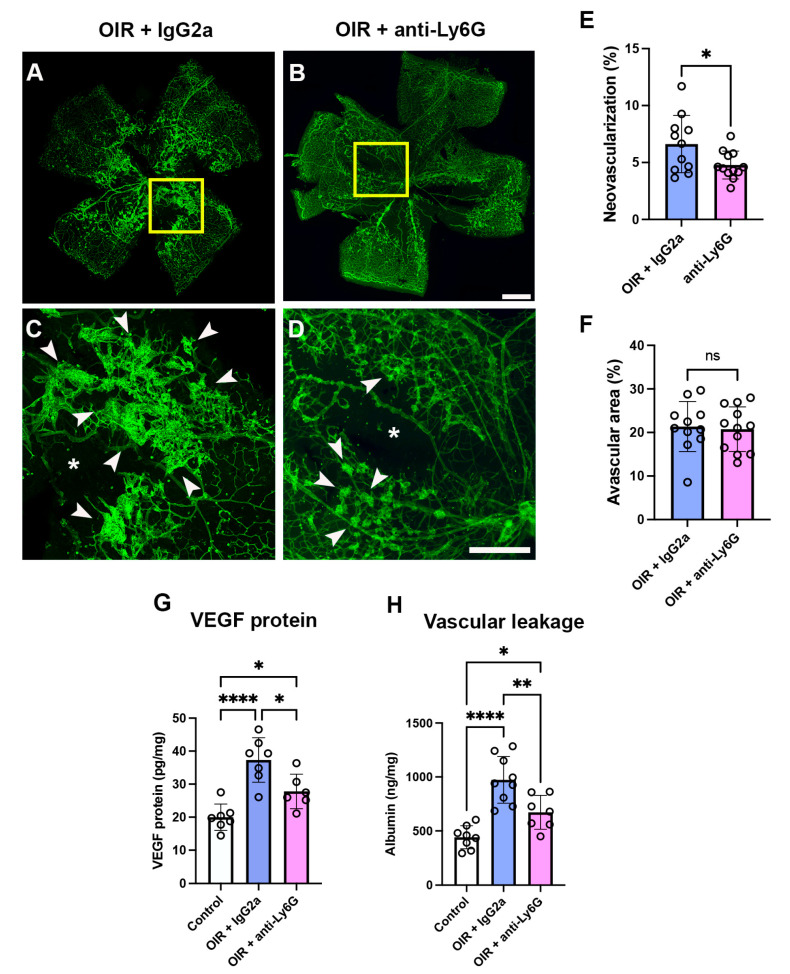
Neutrophil depletion reduced retinal vasculopathy in C57BL/6J mice in phase II OIR (**A**–**D**). Representative retinal flat-mount images from OIR mice at P18, stained with fluorescein isothiocyanate-conjugated isolectin (green) showing neovascularization (arrows). The tissue within the yellow boxes is magnified in (**C**,**D**). Neovascularization (arrows) and vaso-obliteration (asterisks). Scale bar = 100 μm. (**E**) Quantification of neovascularization. (**F**) Quantitation of vaso-obliteration. * *p* < 0.05 according to unpaired *t*-test. ns, non-significant. *n* = 11 to 12 mice per group from three independent experiments. (**G**) Retinal VEGF protein levels via ELISA. (**H**) Retinal vascular leakage via albumin ELISA. * *p* < 0.05, ** *p* < 0.01, **** *p* < 0.0001 according to one-way ANOVA followed by Tukey’s multiple comparisons test. *n* = 6 to 8 mice per group. All values are mean ± SEM.

## Data Availability

The data presented in this study are available on request from the corresponding author.

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
