# Peer review of "Early Depletion of Neutrophils Reduces Retinal Inflammation and Neovascularization in Mice with Oxygen-Induced Retinopathy"

_ijms, 2023, doi:10.3390/ijms242115680_

Round 1

Reviewer 1 Report

Comments and Suggestions for Authors

This manuscript by Suphapimol  V et al., presented the data on the role of neutrophils in retinal vasculopathy. By flow cytometry analysis of blood and spleen, authors found that neutrophil number was increased in at P9 of phase I. Conversely, neutrophil depletion resulted in reduced in vaso-obliteration. And also, authors found that neutrophil depletion approach showed reduced MPO, and vegf protein levels, and reduced transcript levels of TNF, ICAM-1, and MCP-1 at protein level.

The manuscript is nicely written and the data is nicely presented. However, these are my concerns below, which may improve the manuscript:

Minor:

Line 341-Please describe 'cells'

Table 1: I suggest moving Table 1 to 'Supplementary data'

Figure 5: Please re-arrange the figure, maybe bring the E on the top of If, and then bring F, G to the side, or some other way!!!and then improve the size of the graphs.

Line 184: please describe what is a vaso-obliteration

Major: Authors established their hypothesis by providing the Neutrophil quantification from blood and spleen. However, it's crucial to check the neutrophil infiltration in eye/retina.

1. Since the microenvironment play a crucial role, I suggest immunofluorescence-histolgy analysis of retina for neutrophils (Ly6G), co-stain with MPO and DAPI. This data can support in multiple ways; 1) quantify the neutrophil infiltration, 2) validation of neutrophil depletion, 3) Since the authors did not sort Neutrophils, co-staining with MPO will helps to support that neutrophils are the major cells of the inflammatory signal.

2. Do authors consider evaluation of the role of NETS? if so, maybe authors could check for CitH3+MPO by histology.

Reviewer 2 Report

Comments and Suggestions for Authors

The authors performed studies using the mouse model of oxygen-induced retinopathy (OIR) to determine the potential role of neutrophils in the OIR-induced retinal inflammation and pathological retinal  neovascularization.  The rationale for the studies is that neutrophils are acute responders to tissue damage. Thus, the investigators assessed whether neutrophil depletion with an anti-Ly6G mAb influenced retinal inflammation and vascular injury. The study showed that neutrophils were increased in blood and spleen of OIR mice during the hyperoxia phase of OIR but not during the hypoxia phase. They also found that neutrophil depletion by treatment of the OIR mice with anti-Ly6G mAb reduced neutrophils in blood and spleen as well as decreasing levels of myeloperoxidase and inflammatory mediators in the OIR retina. These changes were accompanied by significant reductions in OIR-induced vaso-obliteration and pathological neovascularization.

The results are of potential interest but there are questions/concerns about the methods and data presentation.

1. The protocol used for the OIR model (hyperoxia for 22 hrs, normoxia for 2 hrs from P7 – P12 followed by normoxia from P12 to P18) differs from the well-established model where mice are maintained in hyperoxia continuously from P7 to P12, switched to normoxia on P12, and sacrificed on P17 for analysis of retinal neovascularization and vaso-obliteration. The authors have used the intermittent normoxia model in their previous publications but it would be helpful to explain their rationale for alternating 22 hours of hyperoxia with 2 hours of normoxia during the 5-day hyperoxia treatment and for collecting retinas on P18 rather than on P17 as is done in the established model.

2. The conclusion that neutrophils were increased in blood and spleen in Phase I but not Phase II of OIR is not justified given that the tissues were collected only at the end of Phase II on P18 when the neovascularization is maximal and begins to resolve. The assay should be performed at P14 or P15, while the neovascularization is in progress.

3. The legend for figure 1 requires correction. Panel B shows gating for flow cytometry. Panel C shows neutrophil frequency in blood and spleen and P9 and P18.

4. The images in Figure 2C are of inadequate quality and should be replaced. Vaso-obliterated areas are not visible.

5. What were the effects of neutrophil depletion on vascular repair during Phase II of OIR? The avascular area should be measured at P18 as well as at P12.

6. The retinas shown in panels 5A and 5B appear to be damaged. Better quality images should be provided. The legend for Figure 5 mentions arrows showing neovascularization, but no arrows are present in the figure. Also, what were the treatment effects on the avascular area at P18?

7. Why were inflammatory factors measured at P18 rather than at earlier time points?

8. The statement about “neutrophil-mediated reduction in VEGF levels (line 208)” is confusing and should be revised. The decrease in VEGF levels followed the neutrophil depletion.

Round 2

Reviewer 2 Report

Comments and Suggestions for Authors

The authors have addressed most of my concerns in the revised manuscriopt. However, while they explained their rationale for their modifications of the standard methods for OIR in their rexponse to the review. They did not add this information to the text as was recommended in the review. This information should be provided in the text. There is an error in Lines 139-141 that should be corrected. The text states," In OIR mice treated with the anti-Ly6G mAb, retinal neovascularization and vaso-obliteration were reduced compared to the OIR + IgG2a group (Figure 5A-FE)." The data show no change in vaso-obliteration. This mistake should be corrected.

Author Response

Thank-you. The information about the OIR model has now been included in Lines 267 to 273.

This methodology complies with our institutional animal ethics committee regulations to expose the litters of mice to a short period of room air each day during Phase I to prevent any respiratory distress that may occur in the mothers. We find there are no differences in the retinal pathology that occurs from this protocol compared to continuous exposure to hyperoxia [23]. Phase II has a slight modification to methodologies where mice are killed on P17 [23]. We find no differences in retinal pathological or physiological angiogenesis between P17 and P18 [7,26].

Our apologies for this mistake. Lines 139 to 141 now reads,

“In OIR mice treated with the anti-Ly6G2a mAb, retinal neovascularization was reduced and retinal vaso-obliteration was unchanged compared to the OIR + IgG2a group (Figure 5A-F).”